# A Comprehensive Retrospective Study on the Mechanisms of Cyclic Mechanical Stretch-Induced Vascular Smooth Muscle Cell Death Underlying Aortic Dissection and Potential Therapeutics for Preventing Acute Aortic Aneurysm and Associated Ruptures

**DOI:** 10.3390/ijms25052544

**Published:** 2024-02-22

**Authors:** Jing Zhao, Masanori Yoshizumi

**Affiliations:** Department of Pharmacology, Nara Medical University School of Medicine, 840 Shijo-Cho, Kashihara 634-8521, Japan; jingzhao@naramed-u.ac.jp

**Keywords:** aortic dissection, hypertension, mechanical stretch, vascular smooth muscle cell, cell death, chemokines, inducible nitric oxide synthases

## Abstract

Acute aortic dissection (AAD) and associated ruptures are the leading causes of death in cardiovascular diseases (CVDs). Hypertension is a prime risk factor for AAD. However, the molecular mechanisms underlying AAD remain poorly understood. We previously reported that cyclic mechanical stretch (CMS) leads to the death of rat aortic smooth muscle cells (RASMCs). This review focuses on the mechanisms of CMS-induced vascular smooth muscle cell (VSMC) death. Moreover, we have also discussed the potential therapeutics for preventing AAD and aneurysm ruptures.

## 1. Introduction

In several developed countries, the aging of population has considerably increased the incidence of atherosclerotic cardiovascular diseases (ACDs), which have become serious health issues that warrant immediate clinical attention [1,2]. Atherosclerosis, a key contributor to this issue, can lead to the thickening of the arterial walls, diminished elasticity, and, ultimately, arterial narrowing or occlusion [3,4,5]. These conditions contribute to the restriction of blood flow, thereby increasing the likelihood of hypertension and aortic dissection [6]. Among ACDs, acute aortic dissection (AAD) and arterial aneurysm rupture are the most prevalent vascular diseases associated with high mortality rates [7,8,9]. Physiologically, AAD commences with an initial and sudden tear in the aortic media, which facilitates pulsatile blood entry into the media and subsequent medial layer separation along the vessel length [10,11]. Several endovascular techniques have been developed to treat AAD [12]. However, the most successful therapeutic outcomes are surgical in nature, rather than drug treatment or pharmacotherapy [9,12]. Therefore, investigating the molecular mechanisms underlying AAD is clinically significant for identifying and establishing effective pharmacotherapies for this disease.

Hypertension reportedly occurs in 65–77% of patients with AAD and is a primary risk factor for cardiovascular diseases (CVDs) [13]. Therefore, hypertension can be implicated in the onset of AAD [14,15]. Abnormal haemodynamic load from hypertension-exposed vascular smooth muscle cells (VSMCs) in the medial layers results in cyclic mechanical stretching (CMS), which causes phenotypic switching, cell death, migration, proliferation, and vascular remodelling [16]. Progressive smooth muscle cell loss has been identified in AAD samples and the disease is characterized by aortic medial degeneration [17,18,19]. In addition, mechanical stretch (MS) has also been reported to induce apoptosis in VSMCs [20,21,22,23].

To elucidate the putative mechanisms underlying VSMC death caused by hypertension-associated MS, several studies have reported that endothelin, reactive oxygen species (ROS), angiotensin II, and nitric oxide [23,24,25,26] are associated with hypertension-mediated vascular remodelling. Disrupted molecular signalling may cause VSMC death and remodelling of vessel walls, which are typified by modified function and morphology [27,28]. Activation of kinases, including mitogen-activated protein kinases (MAPKs), Rho GTPases, protein kinase C, and phosphatidylinositol-3-kinase (PI3K)/Akt, has been implicated in MS-mediated cell responses [29,30,31,32]. However, the mechanosensitive cell mechanisms and mechanobiological processes that cause VSMC death remain unclear.

We systematically analysed CMS, which mimics acute blood pressure (BP) increase in rat aortic smooth muscle cells (RASMCs), and examined the factors influencing progression of AAD in our previous study to clarify the possible CMS-mediated cell death mechanisms [33]. We also examined the effects of different clinical drugs on CMS-induced VSMC death [33,34]. Importantly, we provided key insights into the development of novel therapies for hypertension-mediated AAD and aneurysm rupture [35,36]. The findings are summarised in the following sections.

## 2. CMS-Induced Cell Death (Including Apoptosis) and Proliferation in VSMCs

Pulsatile BP generates haemodynamic stimuli in the form of CMS, which acts on blood vessel wall components. VSMCs are particularly vulnerable to CMS resulting from pulsatile BP [37]. Under normal physiological conditions, CMS maintains normal and healthy vasculature by modulating endothelial and VSMC functions. Any changes in the mechanical forces due to arterial disorders may result in VSMC modifications [38,39]. MS is thus considered a hallmark of arterial hypertension as a rise in BP causes an increase in the extent of stretching, which may result in chronic remodelling of the arterial wall and stiffness of the vascular tissue [40]. Abnormal haemodynamic loads caused by hypertension are responsible for AAD and aneurysm rupture [17,18]. Apoptosis and cell death have been identified as critical factors in the development of CVDs [41,42]. Numerous pathophysiological studies have demonstrated that the number of VSMCs markedly decreases with AAD onset, suggesting that VSMC death/apoptosis likely contributes to AAD [43,44,45].

To explore the correlation between VSMC death and AAD/aneurysm rupture, numerous studies have investigated the effects of CMS on VSMC activity and gene expression using in vitro technologies, wherein CMS is applied to cultured cells or tissues to mimic in vivo wall distension [20,21,22,23,46,47,48,49,50,51,52,53,54,55,56,57,58,59]. The majority of such experiments available in the literature have used rats [20,21,23,46,47,48,49,50,51], mice [21,52,53,54,55,56], and human SMCs [21,22,57,58], while a few of them have employed porcine SMCs [59]. It has been reported that applying ≥10% stretching to mouse/rat SMCs cultured on collagen for 4–24 h resulted in enhanced apoptosis in comparison to static controls [20,23,46,47,48,49,51,56]. It has also been reported that stretching human aortic SMCs by >15% (15–25%) yields comparable results [21,22,57,58]. Despite the different experimental conditions and apparatus (uniaxial or equibiaxial strain), these findings indicate that abnormal mechanical stretching induces apoptosis/cell death, regardless of the stretch strength, duration, or SMC origin (Table 1).

Our studies evaluated bio-MS, which mimics acute BP surges, using an experimental stretching load apparatus (STREX^®^, uniaxial model) on in vitro RASMC death [33,34]. This system can create a homogenous CMS on a silicon membrane. RASMCs were produced, cultured, and exposed to CMS for 4 h (60 cycles/min, 15% elongation), as described previously [33]. In these studies, we chose a 4 h duration as the time point at which cell death was initiated in RASMCs. We observed that CMS induced RASMC death in a time-dependent manner (Figure 1), which was consistent with other studies that showed that stretching loads induced smooth muscle cell death [20,49].

In addition, some studies reported that mechanical force initiates signalling pathways which leads to vascular cell death and inflammatory responses, followed by VSMC proliferation [60]. These findings imply that hypertension- or hypertension-associated MS has a two-sided effect on cell death. Various results have been obtained by in vitro and in vivo stretching experiments. The application of physiological stretching (≤10%) to rat aortic vascular SMC for extended periods (48 h to 5 days) can reportedly reduce their proliferation [61,62]. In addition to normal physiological stretching, multiple studies evaluated the effects of hypertension-associated stretching (>10%) on the proliferation of human, mouse, and rat aortic SMCs. Mouse SMCs subjected to 10% stretching for 1 h have been reported to exhibit increased apoptosis and proliferation [52,53,54,55]. Exposing human aortic SMCs to >10% stretching for 12–24 h has been reported to result in enhanced apoptosis and proliferation in comparison to static cells (control) [57,58]. The combined results suggest that normal physiological stretching (<10%) is inactive in SMCs, whereas pathological or hypertension-associated stretching (10–20%) may induce SMC apoptosis and proliferation (Table 2).

We also observed cell proliferation upon application of 15% stretching to RASMCs in vitro for 24 h using a uniaxial stretching apparatus [34]. This could be explained as follows: cell death occurred from the beginning of CMS stimulation up to 4 h, after which the surviving cells entered a proliferation cycle, leading to a progressive increase in cell number due to growth and division. The intensity and duration of CMS stimulation seemed to determine whether the cells died or proliferated. Further studies on the fate of VSMCs under acute CMS conditions are expected to shed light on the molecular mechanisms underlying AAD.

## 3. CMS-Induced MAPK Activation in RASMCs

MAPKs, including three major groups, extracellular signal-regulated kinase 1/2 (ERK1/2), c-Jun N-terminal kinase (JNK), and p38 MAPK (p38), are a family of serine–threonine protein kinases that have emerged as essential components for regulating a variety of cellular responses correlated with mechano-transduction [63]. Among the MAPKs, JNK and p38 are associated with cell death or apoptosis [64].

Previous studies have reported that JNK is phosphorylated in the arterial walls after balloon injury or angioplasty in animal models [65,66] and that it is rapidly activated by cyclic strain stress in VSMCs [59]. JNK has also been found to be highly phosphorylated in the AAD tissues; moreover, it has also been delineated that apoptosis occurs in medial SMC layers [67,68]. Furthermore, stretching stimuli causes p38 phosphorylation in VSMCs [67,69], and the inhibition of JNK phosphorylation leads to regression of AAD [70]. Therefore, we assessed the effects of CMS on JNK and p38 activation (Figure 2) to understand the fundamental signalling processes involved in stretch-induced VSMC death [33,34]. Both JNK and p38 are phosphorylated by CMS, suggesting that their activation is likely involved in CMS-induced RASMC death. Similar results have been reported elsewhere [21,22,47,49,52,53,54,55,57,59]. In addition to JNK and p38, we discovered that ERK1/2, which is commonly implicated in signalling pathways leading to cell proliferation and differentiation, is also phosphorylated by CMS [53]. Despite this, ERK1/2 phosphorylation/activation may not be involved in the signalling pathways that trigger RASMC death because we found that the ERK inhibitors failed to inhibit CMS-induced RASMC death [35]. Taken together, we suggest that CMS causes RASMCs to die and the subsequent onset of AAD via p38- or JNK-mediated intracellular signalling rather than the ERK1/2 signalling pathways.

## 4. Azelnidipine (CS905) (CS) and Olmesartan (Olm) Protect CMS-Induced RASMC Death

We focused on several clinical pharmaceutical products to identify the drugs which inhibit the CMS-induced death of VSMCs. CS is a calcium channel blocker used extensively to treat hypertension, and its efficacy has been primarily ascribed to protect cardiorenal functions by lowering BP [71,72]. In addition to its antihypertensive effects, recent studies have revealed that CS inhibits the growth of aortic aneurysms in mouse models through both anti-inflammatory and antioxidant pathways [73]. Therefore, we examined its effects on CMS-induced RASMC death [33]. As shown in Figure 3, CS, SB203580 (SB), and SP600125 (SP) (p38 and JNK inhibitors, respectively) distinctly increased the viability of stretched RASMC in comparison to unstretched cells (control), indicating that CS and MAPK inhibitors inhibited CMS-induced RASMC death. In addition, CS significantly attenuated CMS-induced JNK and p38 phosphorylation in RASMCs. Similar results were recorded for the SB and SP MAPK inhibitors. Therefore, the abovementioned data suggest that CS prevented CMS-induced RASMC death by inhibiting CMS-induced JNK and p38 phosphorylation.

ROS plays a role in hypertension-mediated vascular remodelling via the activation of matrix metalloproteinases (MMPs) [74]. The mechanical stretching of wild-type VSMCs has been reported to result in rapid ROS formation [75]. Mechanical stress can also cause oxidative stress and mitochondrial malfunction, resulting in the production of pro-apoptotic substances, such as cytochrome c, which activates caspases and causes cell death [76]. We previously reported that p38 and JNK are vulnerable to oxidative stress [77]. CS has been shown to possess anti-oxidant effects in mouse aneurysmal models [73,78]. Therefore, it seems reasonable to speculate that CS can prevent JNK and p38 activation via antioxidative processes, which in turn can prevent CMS-induced cell death. To verify the accuracy of this hypothesis, we evaluated the effects of antioxidants, diphenyleneiodonium (DPI) and tempol, on CMS-induced RASMC death [33]. Our experiments demonstrated that both DPI and tempol pretreatment failed to inhibit stretch-induced RASMC death, indicating that oxidative stress might not be a dominant factor influencing CMS-induced RASMC death, at least under our experimental conditions.

Angiotensin II plays a role in the aetiology of aneurysms by activating the MAPK pathway [79]. Recent studies have reported that the angiotensin II receptor (G protein-coupled AT1 receptor) is activated by CMS via agonist-independent mechanisms [80]. It has been reported that, in the absence of agonist stimulation in cultured cells, the angiotensin II receptors are activated via the receptor initiated intracellular signalling cascades in response to CMS. Furthermore, stretch-induced AT1 receptor activation has been observed in the mesenteric and renal arteries of angiotensinogen-knockout animals [81].

Our investigations showed that mechanical stretch, which is unrelated to angiotensin II stimulation, phosphorylated JNK along with p38 and caused SMC death. Although we did not evaluate the concentration of angiotensin II in the medium, it is unlikely to be implicated in JNK and p38 phosphorylation. Therefore, it is plausible that AAD onset may be influenced and triggered not only by agonist stimulation but also by mechanical stretch.

Angiotensin II receptor blockers (ARBs) are a group of medications used to treat hypertension by inhibiting the renin–angiotensin system [82]. However, in addition to their hypotensive actions, ARBs also have organ-protecting benefits that contribute to their effectiveness. A particular type of ARB blocks both agonist- and stretch-induced activation [83]. Olm is a powerful ARB with inverse agonist functions [84] which can prevent basic and stretch-induced activation of the AT1 receptor [83,84,85]. We previously reported that Olm inhibits RASMC migration by inhibiting JNK activation [86]. Further research has reported the effects of Olm on CMS-induced cell death in RASMCs and the associated alterations in stretch-induced intracellular signalling (e.g., JNK and p38), with a particular focus on Olm [34]. Olm, SB, and SP MAPK inhibitors greatly improved the viability of stretched RASMCs (Figure 3). Similar to CS, Olm also inhibited CMS-induced JNK and p38 phosphorylation in RASMCs, suggesting that Olm prevents CMS-induced RASMC death by inhibiting p38- or JNK-mediated intracellular signalling. Interestingly, candesartan, another well-known inverse agonist of AT1 receptor, has also been found to reduce stretch-induced cardiac hypertrophy [86], which is in agreement with our findings.

From the above results, it is clear that both CS and Olm suppress JNK and p38 phosphorylation along with CMS-induced RASMC death, independent of their hypotensive actions. It should be noted that the experiments described above were carried out at a fixed CMS frequency of 60 cycles/min (1 Hz), neglecting the impact of variable CMS frequencies on VSMCs. β-blockers have long been used as a mainstay of AAD pharmacologic treatment to reduce both blood pressure and heart rate [87]. Therefore, in addition to other calcium channel blockers and angiotensin II receptor antagonists, future studies should also focus on β-blockers and compare their effects on stretch-induced RASMC death.

## 5. Exploring CMS-Induced Cell Death Mechanisms

The abovementioned results demonstrate that hypertension-associated MS induces VSMC death, which is thought to cause alterations in the extracellular matrix, thus weakening the vessel wall and leading to the development of CVDs [43]. It is considered that excessive CMS disrupts the extracellular matrix surrounding VSMCs, leading to structural support loss and subsequent cell death [88]. Furthermore, autophagy induced by CMS triggers cell death through the degradation and recycling of cellular components [89]. In addition, abnormal CMS stimulation causes pro-inflammatory molecules to be released in the blood vessel wall, potentially contributing to VSMC death and the progression of CVDs [31,70]. Although there is abundant evidence indicating its crucial role in the onset of AAD, the mechanisms by which mechanical stress transmits signals to induce VSMC death remain unclear.

### 5.1. Various Mechanisms Underlying VSMC Death Induced by Bio-MS

Recent studies have revealed multiple putative signalling mechanisms underlying MS-induced VSMC death. It has been reported that applying periodic stretching (20% elongation, 0.5 Hz, 6 h) to in vitro RASMCs induces VSMC apoptosis through endothelin B receptor-mediated mechanisms [23]. This study indicated that the endothelin system plays a key role in the initiation of hypertension-induced remodelling in conduit arteries, which may begin with stretch-induced VSMC death. Wernig et al. demonstrated that MS-induced apoptosis in VSMCs is mediated by beta1-integrin-rac-p38-p53 signalling pathways [20,21]. Integrin signalling, and not the growth factor receptor-ERK pathways, have been implicated in signal transduction leading to stretch-induced p53 expression.

Other studies have shown that MS-induced p38 activation regulates acute cell apoptosis after vascular injury and plays a role in transducing signals that lead to venous bypass graft atherosclerosis [55]. Moreover, CMS increases the expression of p53-up-regulated modulator of apoptosis (PUMA) in cultured human VSMCs via the interferon-gamma (IFN-γ), JNK, and interferon regulatory factor-1 (IRF-1) pathways [22]. According to this study, PUMA plays a critical role as a mediator of the CMS-induced death of VSMCs. In another study, when a uniform equibiaxial cyclic strain was applied to VSMCs in vitro, CMS inhibited VSMC growth while promoting VSMC apoptosis, partially by modulating the Notch receptor and downstream target-gene expression [49]. Additionally, CMS stimulation reduces Hedgehog expression in VSMCs both in vitro and in vivo, which correlates with a large increase in VSMC death [46].

Mechanical injury-induced vascular remodelling with activated SMC migration can be accelerated by apoptosis signal-regulating kinase 1 (a MAPK kinase) through enhanced neovascularization and/or increased VSMC and endothelial cell apoptosis [90]. Furthermore, endoplasmic reticulum (ER) stress in response to CMS has been shown to promote VSMC apoptosis and degeneration, contributing to the formation and progression of thoracic AAD [91]. In vitro studies using rat thoracic VSMCs have shown that CMS greatly induces the apoptosis of VSMCs and the formation of AAD, which can be suppressed by the upregulation of Yes-associated protein 1 (YAP1) via the Hippo-YAP signalling pathway [51]. Another study indicated that ER stress can alter intracellular YAP1 protein expression in VSMCs, and that YAP1 provides protection against ER-induced VSMC apoptosis by blocking caspase-8/3 activation, which is partly mediated by the upregulation of ANKRD1 [92]. In addition, MS induces both cell proliferation and death through PDI/NOX1/ROS signalling, and VSMC apoptosis relies on the caspase-3 signalling pathway [54]. However, the involvement of other caspases in MS-induced VSMC death remains unclear. Other studies have reported that miRNA upregulation by CMS may modulate the activation of numerous downstream transcription factors and ultimately affect VSMC death [48,57,58]. According to these studies, miR-124-3p inhibits Lamin A/C, which increases VSMC death under cyclic strain [48]; miR-421 modifies the renin–angiotensin system and ACE2 protein levels, potentially influencing both apoptosis and hypertension [57]; moreover, increased miR-21 expression may prevent stretch-induced apoptosis in HASMCs [58]. These findings imply that, in response to mechanical stretch, miRNAs, such as miR-124-3p, miR-421, and miR-21, also play a crucial role in apoptosis and cellular dysfunction. Despite the progress made in this field, mechanosensitive cell mechanisms and mechanobiological processes are not completely understood. Further studies on molecular signalling of bio-MS and related alterations in gene expression that contribute to AAD progression are warranted.

### 5.2. Research on CMS-Induced Cell Death Mechanisms in Our Laboratory

CS and Olm have been extensively studied as antihypertensive agents. However, the majority of the investigations have only examined their protective effects towards cardiorenal functions by lowering BP [71,72,79]. In our studies, CMS stimulation was conducted by mimicking acute BP surges in an in vitro experimental stretching load apparatus, and when Olm or CS were added, the simulated BP increases remained unaffected. Therefore, it can be inferred that other pathophysiological mechanisms exist via which CS or Olm protect VSMCs against death/apoptosis and result in subsequent AAD progression, independent of their BP-lowering effects. To identify these potential mechanisms, we screened stretch-induced RNA expression levels in RASMCs (cultured using collagen I membranes and exposed to 15% stretch for 4 h) and performed cDNA microarray and bioinformatics analyses. These investigations identified 91 differentially expressed genes (DEGs) in CMS-exposed RASMCs, of which 29 were putatively regulated by p38 or JNK and associated with cell death [35]. Using real-time polymerase chain reaction studies to evaluate DEGs in stretched RASMCs, we observed (for the first time) substantial transcriptional changes in DEGs implicated in inflammation in response to CMS, including inducible nitric oxide synthase (iNOS) and other chemokines (Figure 4). Among the inflammation-related DEGs, the expression of C-X-C motif chemokine ligand 1 (Cxcl1) and C-X3-C motif chemokine ligand 1 (Cx3cl1) was induced by CMS in a JNK-dependent manner, with Cxcl1 expression increasing more dramatically than Cx3cl1 expression. On the other hand, iNOS expression was induced in a p38-dependant manner [31]. These results were obtained by assessing the expression of chemokines and iNOS in RASMCs treated with JNK and p38 inhibitors. Cxcl1 and Cx3cl1 transcript levels induced by CMS were decreased by a JNK inhibitor (SP600125), suggesting that their gene expression in the presence of CMS was transcriptionally dominated by JNK. In contrast, the p38 inhibitor (SB203580) had no effect on Cxcl1 or Cx3cl1 expression. However, it reduced iNOS expression and NO production in RASMCs subjected to CMS. This indicates that p38 is involved in CMS-mediated iNOS-NO signalling. Although the mechanisms involved in transcript induction by CMS appear to differ depending on the cell type, the expression of these genes may play crucial roles in stress response to hypertension-associated CMS.

It is generally recognized that bio-MS can induce the release of pro-inflammatory molecules in the blood vessel wall, potentially leading to VSMC death and the progression of vascular diseases [22,47,50,59]. The nuclear factor-kappaB (NF-κB) pathway can activate the transcription of pro-inflammatory cytokines, such as tumour necrosis factor-alpha (TNF-α) and interleukin-6 (IL-6), which can also promote apoptosis in VSMCs [93]. Considering that JNK induces the expression of multiple inflammatory regulators (e.g., iNOS and IL-1a) during aneurysm growth [71] and that NF-κB controls gene expression during inflammatory responses [93], we analysed chemokine and iNOS expression in association with NF-κB activation. Cx3cl1 transcript levels, with and without CMS, decreased due to a selective NF-κB inhibitor (BAY11-7082), suggesting that consistent Cx3cl1 expression was regulated in a NF-κB-dependent fashion. In contrast, Cxcl1 and iNOS transcript levels after CMS slightly changed due to the NF-κB inhibitor, suggesting that Cxcl1 and iNOS induction by CMS is putatively NF-κB independent. These data indicate that Cxcl1 and iNOS do not induce or participate in inflammatory responses, at least during the initial stages of CMS-induced RASMC death [35,36].

Additionally, we observed that antagonists of chemokine receptors increased the death of stretched RASMCs, suggesting that both Cxcl1 and Cx3cl1 protected RASMCs from CMS-induced cell death in the early stages. In contrast, Cxcl1 and Cx3cl1 have been reported to promote atherosclerosis [94]. Cxcl1 released by endothelial cells promotes arterial inflammation by acting as a neutrophil chemoattractant [95], while Cx3cl1 suppresses monocyte and foam cell death in atheromatous plaques [96], both of which contribute to vascular remodelling and subsequent atherogenesis. Polymorphisms in the Cx3cl1 receptor, Cx3CR1, have been identified as genetic risk factors for atherosclerosis [97], and the deletion of Cx3cl1, Cx3CR1 and Cxcl1 suppresses atherogenesis [98,99]. Considering that chronic chemokine use may aggravate vascular remodelling by triggering inflammation, our findings suggest that acute chemokine use could protect VSMCs from hypertension-induced cell death, which might, in turn, prevent vascular remodelling and AAC onset.

We also examined the effects of CMS-induced iNOS expression and NO production on RASMC death using an iNOS inhibitor (1400 W) [36]. RASMC death was found to increase substantially with increasing iNOS inhibitor concentrations and decrease with increasing NO donor DETA-NONOate concentrations. The amount of NO produced by iNOS was highly dependent on the enzyme’s expression levels. NO is regarded as a major and potent mediator of vasodilation. Low amounts of NO generated in cells promote cardiovascular homeostasis, whereas excessive NO levels may be deleterious to the cardiovascular system and contribute to hypertension [100]. Thus, we suggest that CMS stimulates iNOS expression and NO production in RASMCs, thereby protecting them from CMS-induced cell death in the early stages.

Bioinformatics analyses (iRegulon) incorporating mRNA data from stretched RASMCs were used to explore the signalling pathways involved in CMS-regulated RASMC death [35]. Analyses showed that a signal transducer and activator of transcription 1 (STAT1) potentially regulated 11 DEGs, including Cxcl1, Cx3cl1, and iNOS. Western blot analysis of RASMCs verified that STAT1 was phosphorylated when the cells were exposed to CMS. However, MAPK inhibitors had little effect on CMS-induced STAT1 phosphorylation, whereas STAT1 inhibitors did not affect MAPK phosphorylation, suggesting that MAPK activation and regulation in stretched VSMCs are distinct from STAT1 activation. Further investigations are required to elucidate the crosstalk between MAPK and STAT1 signalling pathways in stretch-induced VSMC death.

To further validate these data, we used an abdominal aortic constriction (AAC) mouse model to examine whether Cxcl1, Cx3cl1, and iNOS expression were induced by AAC-related hypertension in vivo. AAC effectively induced sustained and elevated mean arterial pressure without vasopressors [101]. Consequently, Cxcl1 and iNOS transcript levels in arteries subjected to AAC for 6 h were higher than those in sham-treated arteries, whereas Cx3cl1 transcript levels exhibited no discernible alterations [35,36]. Immunofluorescence analyses using antibodies against Cxcl1, Cx3cl1, and iNOS revealed that Cxcl1 and iNOS signals were considerably enhanced in the arteries subjected to AAC in comparison to the sham-treated arteries; however, no obvious changes in Cx3cl1 signals were observed. These data support our findings regarding CMS-stretched RASMCs in vitro, indicating that Cxcl1 and iNOS expression in VSMCs is induced during the early stages of hypertension.

## 6. Conclusions

AAD is a serious vascular disorder which manifests as sudden severe pain and has a high fatality rate in the clinical settings. Regardless of surgical treatments, the current pharmacotherapies for AAD mainly rely on antihypertensive medications that protect cardiorenal functions by reducing BP. In addition to this therapy, novel therapeutics must be explored and identified to generate more effective pharmacotherapies for preventing this disease. Numerous efforts have been made to clarify the potential mechanisms underlying VSMC death/apoptosis caused by hypertension-associated MS with the aim to implement the development. And various factors and signalling pathways triggering VSMC death/apoptosis have been revealed, though there are still some discrepancies in the results which are probably due to different experimental conditions and/or approaches.

In the systematic studies we carried out, we hypothesised that acute BP elevation increases bio-MS on the arterial walls and induces cell death, thereby causing AAD onset. Our experimental data demonstrated that CMS induced death of RASMCs via p38- or JNK-mediated intracellular signalling pathways. Simultaneously, we found that CMS increased iNOS expression and NO synthesis via p38 or STAT1 activation, or induced the expression of Cxcl1 and Cx3cl1 via the JNK-mediated cellular signalling pathway, all of which protected RASMCs from CMS-induced cell death. The results suggest that chemokines and iNOS may play critical roles in the pathological progression of AAD. In addition, we examined the effects of clinical pharmaceutical products such as CS and Olm on CMS-induced VSMC death and found that both CS and Olm prevented CMS-induced RASMC death by inhibiting p38 or JNK phosphorylation, independent of their hypotensive actions. Therefore, targeting MAPKs, STAT1, and their downstream transcription factors (Cxcl1, Cx3cl1, and iNOS) could provide a potential therapeutic strategy for preventing CMS-induced VSMC death and subsequent AAD progression (Figure 5). Research is on-going to validate these targets in in vivo models, explore the mechanisms by which CMS transmits signals to induce AAD onset, and identify novel therapeutics for the treatment of AAD.

## Figures and Tables

**Figure 1 ijms-25-02544-f001:**
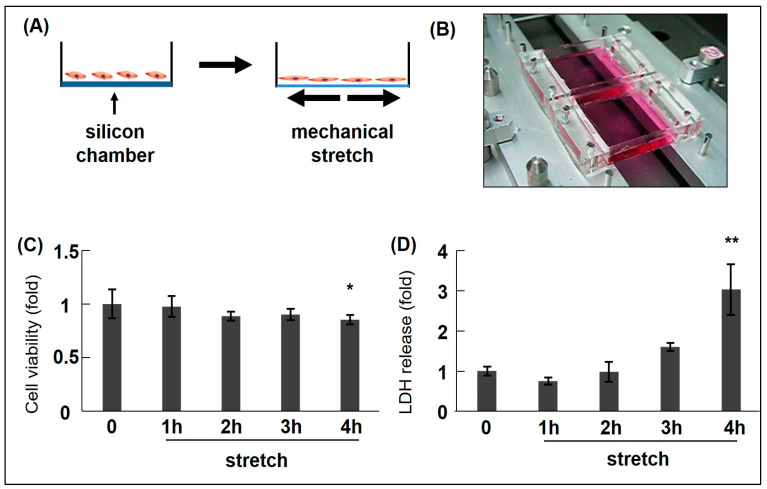
Cyclic mechanical stretch (CMS) schematic/apparatus and CMS-induced rat aortic smooth muscle cell (RASMC) viability/death data. (**A**) Schematic describing CMS (mimicking hypertension). (**B**) The mechanical stretch apparatus (Model STB-1400, STREX Co., Ltd., Osaka, Japan). RASMCs were subjected to CMS (15% elongation) for different time intervals (0–4 h) and incubated for 1 day. Cell death/viability levels were examined using 3-(4,5-dimethylthiazol-2-yl)-2,5-diphenyl-2H-tetrazolium bromide (MTT) (**C**) and lactate dehydrogenase (LDH) release assays (**D**). Colorimetric data were normalized to control data by arbitrarily setting their absorbance values (0 h) to 1. Data are indicated by mean ± standard deviation (n = 4). Asterisks refer to significant differences when compared with controls (* *p* < 0.05, ** *p* < 0.01). Modified from Zhao et al. [33].

**Figure 2 ijms-25-02544-f002:**
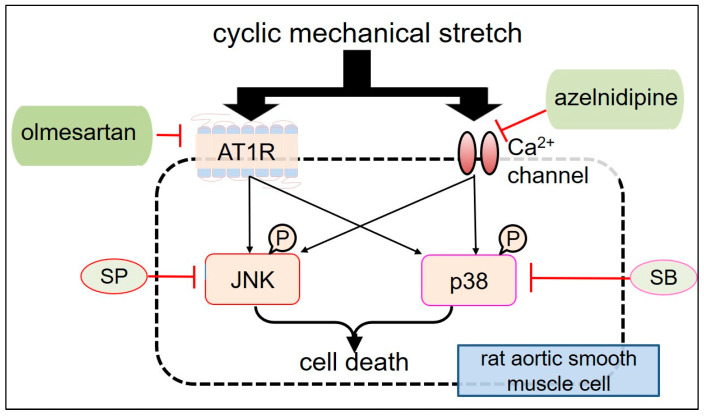
Schematic showing azelnidipine and olmesartan mediated-effects on CMS-induced JNK and p38 activation in RASMCs. AT1R: Angiotensin II receptor type 1; SP600125 (SP): JNK inhibitor; SB203580 (SB): p38 inhibitor; RASMC: rat aortic smooth muscle cell.

**Figure 3 ijms-25-02544-f003:**
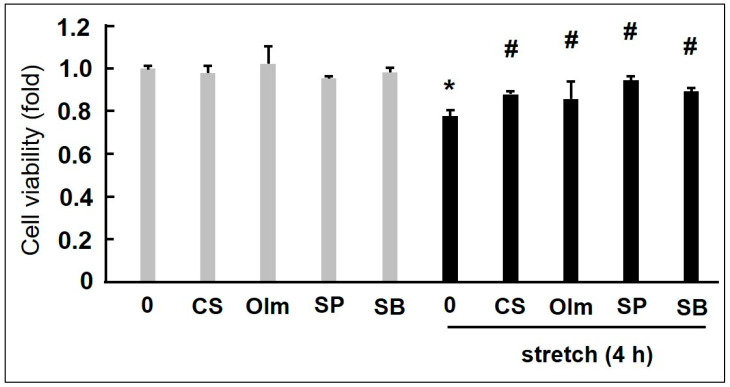
CMS-induced-RASMC viability data plus azelnidipine (CS905, CS), olmesartan (Olm), and MAPK inhibitors (SP and SB). RASMCs pre-incubated with Olm, CS, SP, and SB for 20 min were exposed to CMS (15% elongation) for 4 h and incubated for 1 day. Cell viability was examined using MTT assays. Colorimetric analysis of each value was normalized by arbitrarily setting the colorimetric value of the control cells without stretch to 1. Data are indicated by mean ± standard deviation (n = 4). (* *p* < 0.05 and # *p* < 0.05 compared with control without stretch and stretch only, respectively). Modified from Zhao et al. [33] and Ito et al. [34].

**Figure 4 ijms-25-02544-f004:**
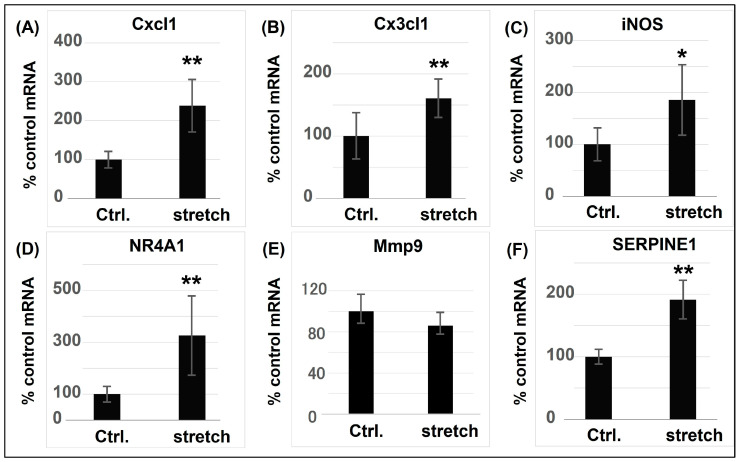
Candidate gene transcript expression in CMS-exposed RASMCs. RASMCs were exposed to CMS for 4 h after which Cxcl1 (**A**), Cx3cl1 (**B**), iNOS (**C**), NR4A1 (**D**), Mmp9 (**E**), and SERPINE1 (**F**) expression transcripts were evaluated (RT-PCR), expressed as a percentage of control (Ctrl.), and normalized to GAPDH. Data indicate the mean ± standard error (n = 6) * *p* < 0.05 and ** *p* < 0.01 versus control. Modified from Zhao et al. [35,36]. NR4A1: nuclear receptor subfamily 4 group A member 1; Mmp9: matrix metalloproteinase-9; SERPINE1: serpin family E member 1; GAPDH: Glyceraldehyde 3-phosphate dehydrogenase.

**Figure 5 ijms-25-02544-f005:**
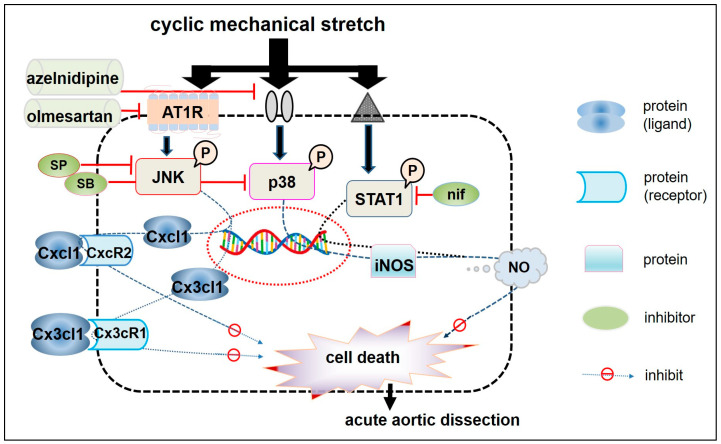
Graphic showing CMS-induced cell death mechanisms in RASMCs and putative drug targets. CMS applied to VSMCs induces JNK, p38 and STAT1 phosphorylation leading to cell death and evokes down-stream multiple signalling pathways non-specifically. Upon the process, Cxcl1 and Cx3cl1 expression is induced in a JNK-dependent manner, while iNOS expression is induced via STAT1 and p38 signal pathways independently. The induced Cxcl1, Cx3cl1, and iNOS play positive roles in protecting VSMCs from CMS-induced cell death. Nifuroxazide (nif): STAT1 inhibitor; NO: Nitric Oxide; CxcR2: C-X-C motif chemokine receptor 2; Cx3cR1: C-X3-C motif chemokine receptor 1.

**Table 1 ijms-25-02544-t001:** Cyclic mechanical stretch (CMS) induced cell death/apoptosis of vascular smooth muscle cells (VSMCs).

Cell Type (ASMC)	Type of Stretch Frequency, Intensity, Duration	Functions	Related Molecules	Study
Rat	uniaxial model, 1 Hz, 15%, 4 h	cell death ↑	iNOS, p38, STAT1	36
Rat	FX-5000T, 1.25 Hz, 15%, 24 h	apoptosis ↑	Lamin A/C, miR-124-3p, TP53, CREB1, MYC, STAT1/5/6, JUN	48
Rat	uniaxial model, 1 Hz, 15%, 4 h	cell death ↑	Cxcl1, Cx3cl1, JNK, STAT1	35
Rat	Flexcell, 1 Hz, 18%, 48 h	apoptosis ↑	YAP1	51
Rat	uniaxial model, 1 Hz, 15%, 4 h	cell death ↑	JNK, p38, ERK1/2	34
Rat	FX-2000, 1 Hz, 10–20%, 6 h up to 18 h	apoptosis ↑	GADD153, TNF-α, JNK, AP-1, caspase-3	47
Rat	FX-4000T, 1 Hz, 0–10%, 0–24 h	apoptosis ↑	Notch 3, sonic hedgehog, patched 1	46
Rat	FX-4000T, 1 Hz, 0–15%, 0–24 h	apoptosis ↑	Notch 3 receptor, MAPK, Gi-protein, caspase-3, Bax, Bcl-xL	49
Rat	FX4000, 1 Hz, 7% or 15%, 6 h	apoptosis ↑	β1-integrin, Rac, p38, p53	20
Rat	FX-3000, 0.5 Hz, 20%, 6 h	apoptosis ↑	ET-1, endothelin B receptor	23
Mouse	FX-5000, 1 Hz, 12%, 24 h	apoptosis ↑	EZH2, YAP, TEAD1	56
Human	FX-2000, 1 Hz, 10–20%, 18 h	apoptosis ↑	PUMA, JNK, IFN-γ, IRF-1	22
Porcine	Equi-biaxial dynamic stretch, 1 Hz, 7% or 25%, 24 h up to 72 h	apoptosis ↑	TNF-α receptor-1, TRAF-2, JNK, p38, TNF-α	59
Rat, Mouse, Human	FX-3000, 1 Hz, 5% to 25%, 10 min up to 6 h	apoptosis ↑	Rac, p38, p53, Bcl-2, Bcl-xL, Bax	21

ASMC: aortic smooth muscle cell; ↑: increased; iNOS: inducible nitric oxide synthases; p38: p38 MAPK; MAPK: mitogen-activated protein kinase; STAT1: signal transducer and activator of transcription 1; miR-124-3p: MicroRNA-124-3p; TP53: tumour protein p53; p53: tumour protein p53; CREB1: CAMP responsive element binding protein 1; MYC: transcript variant 2; JUN: transcription factor Jun; Cxcl1: C-X-C motif chemokine ligand 1; Cx3cl1: C-X3-C motif chemokine ligand 1; JNK: c-Jun N-terminal kinase; YAP: yes-associated protein 1; ERK1/2: extracellular signal-regulated kinase 1/2; GADD153: DNA damage-inducible transcript 3 protein; TNF-α: tumour necrosis factor-α; AP-1: activator protein-1; Notch 3: neurogenic locus notch homolog protein 3; Bax: Bcl-2-associated X protein; Bcl-2: B-cell lymphoma 2; Bcl-xL: Bcl-extra large; Rac: RAS-related C3 botulinus toxin substrate; ET-1: endothelin-1; EZH2: enhancer of zeste homolog 2; TEAD1: transcription factor interacting domain 1; PUMA: p53-up-regulated modulator of apoptosis; IFN-γ: interferon-γ; IRF-1: interferon regulatory factor-1; TRAF-2: TNF receptor-associated factor 2.

**Table 2 ijms-25-02544-t002:** CMS regulated apoptosis and proliferation of VSMCs.

Cell Type (ASMC)	Type of Stretch Frequency, Intensity, Duration	Functions	Related Molecules	Study
Rat	FX-5000T, 1 Hz, 15%, 24 h FX-5000T, 1 Hz 10% or 20%, 24 h	apoptosis ↓, proliferation ↑ apoptosis ↑, proliferation ↓	YAP	50
Mouse	FX-3000, 1 Hz, 10%, 1 h	apoptosis ↑, proliferation ↑	PDI, ERK, JNK, p38, caspase-3/12	52
Mouse	FX-3000, 1 Hz, 0–10%, 1 h	apoptosis ↑, proliferation ↑	ERK, JNK, p38	53
Mouse	FX-3000, 1 Hz, 10%, 1 h	apoptosis ↑, proliferation ↑	ERK, JNK, p38, NF-κB/p65, caspase-3	55
Mouse	FX-3000, 1 Hz, 10%, 1 h	apoptosis ↑, proliferation ↑	PDI, NOX1, ROS, caspase-3	54
Human	FX-5000T, 1 Hz,18%, 12 h	apoptosis ↑, proliferation ↑	p38, ATF3, ACE2, miR-421	57
Human	FX-5000, 1 Hz, 10–16%, 12 h	apoptosis ↑, proliferation ↑	miR-21, NF-κB, AP-1, PDCD4	58

↑: increased; ↓: decreased; PDI: protein disulfide isomerase; NF-κB: nuclear factor-kappa B; NOX1: NADPH oxidase 1; ROS: reactive oxygen species; ATF3: activating transcription factor 3; ACE2: angiotensin-converting enzyme 2; miR-421: MicroRNA-421; miR-21: MicroRNA-21; PDCD4: programmed cell death 4.

## Data Availability

No data have been generated in this study.

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
