# Peer review of "A Comprehensive Retrospective Study on the Mechanisms of Cyclic Mechanical Stretch-Induced Vascular Smooth Muscle Cell Death Underlying Aortic Dissection and Potential Therapeutics for Preventing Acute Aortic Aneurysm and Associated Ruptures"

_ijms, 2024, doi:10.3390/ijms25052544_

Round 1

Reviewer 1 Report

Comments and Suggestions for Authors

Authors proposed a paper entitled " A comprehensive retrospect on mechanisms of cyclic mechanical stretch-induced vascular smooth muscle cell death underlying aortic dissection and potential therapeutics for preventing acute aortic aneurysm and associated ruptures" as a review referring to mechanisms underlying VSMC death caused by cyclic mechanical stretching and novel therapies used to block cells death.

The review is well written and documented and the paper deserves to be published.

I could suggest several minor points to further improve the review:

1. In section 2, that refers to the stretching studies on mice and rats' SMCs, please specify these results also, or unite the paragraphs that mention all 4 experimental models. Please clarify the results that refer to the physiological conditions (<10% stretching) and the pathological ones (10-20% stretching).

2. Are there any references to pyroptosis as a cell death mechanism in SMCs after exposure to cyclic mechanical stretch?

3. In section 5.1, lines 293-295, please mention some miRNA examples.

4. In line 340 there is a missing space before the word "Considering".

5. Please rewrite the conclusion section as this section should not contain new additional information found only in the main text. All new information and references should be included in the main text.

Author Response

List of revisions and discussion

We would like to express our appreciation to the reviewers and editors for their good suggestions and comments. These are very valuable for us to revise our manuscript. Please find below the response to the Reviewer.

Comments-1: In section 2, that refers to the stretching studies on mice and rats' SMCs, please specify these results also, or unite the paragraphs that mention all 4 experimental models. Please clarify the results that refer to the physiological conditions (<10% stretching) and the pathological ones (10-20% stretching).

Response 1:

It is a good suggestion. According to this suggestion, we divided the original table (Table1) into two tables (Table 1 and 2) in terms of the observed results regarding cell death and proliferation. Furthermore, in each table, we categorized the results based on the experimental models (i.e., rat, mouse, human, and porcine) and clarify the results for each species under physiological (<10% stretching) and pathological (10-20% stretching) conditions. Please refer to the revised manuscript with the updated results tables (Table 1 and Table 2).

Comments-2: Are there any references to pyroptosis as a cell death mechanism in SMCs after exposure to cyclic mechanical stretch?

Response 2:

Current research shows a correlation between cell pyroptosis and aortic dissection, but there is still inconclusive evidence connecting VSMC death to pyroptosis after exposure to cyclic mechanical stretch. For the reason, pyroptosis is not addressed in the manuscript.

No doubt, pyroptosis is an interesting area of research, we will pay attention to the relevant studies in future.

Comments-3: In section 5.1, lines 293-295, please mention some miRNA examples.

Response 3:

According to the suggestion, we have added three miRNA examples. Please refer to lines 292-298 in the revised manuscript.

Comments-4: In line 340 there is a missing space before the word "Considering".

Response 4:

The missing space has been added (line 343).

Comments-5: Please rewrite the conclusion section as this section should not contain new additional information found only in the main text. All new information and references should be included in the main text.

Response 5:

According to the suggestion, we have re-written the conclusion section. Please refer to lines 405-409 and lines 413-421 in the revised manuscript.

Reviewer 2 Report

Comments and Suggestions for Authors

Very well described and well presented study with experimental evidence supporting the ideas and hypothesis.

What remains unclear is the relevance of this analysis with regards to current therapy for acute aortic dissection (AAD). Provide a justification to establish the therapeutic relevance of the proposed ideas in comparison to ongoing therapies for AAD

Comments on the Quality of English Language

none.

Author Response

We would like to express our appreciation to the reviewers and editors for their good suggestions and comments. These are very valuable for us to revise our manuscript. Please find below the response to the Reviewer.

Comments-1: What remains unclear is the relevance of this analysis with regards to current therapy for acute aortic dissection (AAD). Provide a justification to establish the therapeutic relevance of the proposed ideas in comparison to ongoing therapies for AAD.

Response 1:

We have rewritten the conclusion section, where we explain the recent findings that azelnidipine and olmesartan protect against mechanically induced vascular smooth muscle cell death independent of their antihypertensive effects. Elucidating the mechanisms by which they regulate MAPKs and downstream iNOS and chemokines may help discover new therapeutic approaches.
